# KoRe: Compact Knowledge Representations for Large Language Models

Davide Cavicchini[1,*], Fausto Giunchiglia[1] and Jacopo Staiano[1]

[1]*University of Trento, Via Calepina, 14, 38122 Trento TN, Italy*

**Abstract**

Modern Large Language Models (LLMs) have shown impressive performances in user-facing tasks such as question answering, as well as consistent improvements in reasoning capabilities. Still, the way these models encode knowledge seems inherently flawed: by design, LLMs encode world-knowledge within their parameters. This way of representing knowledge is inherently opaque, difficult to debug and update, and prone to hallucinations. On the other hand, Knowledge Graphs can provide human-readable and easily editable world knowledge representations, and their application in knowledge-intensive tasks has consistently proven beneficial to downstream performance. Nonetheless, current integration techniques require extensive retraining or finetuning. To overcome this issue, we introduce KoRe, a methodology to encode 1-hop sub-graphs into compact discrete knowledge tokens and inject them into a LLM backbone. We test the proposed approach on three established benchmarks, and report competitive performances coupled with a significant reduction (up to 10x) in token usage. Our results show that compact discrete KG representations can efficiently and effectively be used to ground modern LLMs.

The code will be available at https://github.com/DavidC001/KoRe

**Keywords**

Large Language Models, Knowledge Graph, Token Embedding, Graph Neural Networks, Vector Quantization

## 1. Introduction

Arguably, Large Language Models have revolutionized natural language processing, demonstrating unprecedented capabilities in text generation, reasoning, and complex task execution. However, these models bear a fundamental limitation, as their knowledge is implicitly encoded within billions of parameters, making it opaque, static, and prone to hallucinations: when dealing with factual information, LLMs may generate plausible-sounding but incorrect responses, struggle with recent events not present in their training data, or fail to provide verifiable sources for their claims.

We posit Knowledge Graphs (KGs) as a valid solution to these challenges [1, 2]. Unlike the parametric knowledge stored in LLMs, KGs provide **structured**, **interpretable**, and **updatable** representations of factual information [3]. KGs organize knowledge as networks of entities connected by typed relationships, enabling precise querying, easy verification, and efficient updates. The central challenge, however, is bridging the representational gap between a graph and the token sequences a language model consumes. This integration often relies on **textual serialization** of KG information, converting structured triples into natural language descriptions that are then concatenated with the input prompt. This approach suffers from severe token inefficiency: a single logical fact may consume dozens of tokens when expressed in natural language, and high computational costs.

Some studies proposed the use of small LoRA-like [4] **adapters** to store new information in the model's parameters, or rely on trained Knowledge Graph Embedding (KGE) models and inject them into inner layers of the model; however, these techniques require re-training for each new piece of information, which can lead to costly updates in dynamic environments. An alternative solution relies on injecting knowledge as **continuous embeddings** rather than text. Prior work has explored this

*GenAIK-NORA 2026: Workshop on Generative AI and Knowledge Graphs & Knowledge Graphs and Agentic Systems Interplay, co-located with IJCAI-ECAI 2026, August 15–17, 2026, Bremen, Germany*

*Corresponding author.

✉ davide.cavicchini@unitn.it (D. Cavicchini); fausto.giunchiglia@unitn.it (F. Giunchiglia); jacopo.staiano@unitn.it (J. Staiano)

🆔 0009-0005-9662-8496 (D. Cavicchini); 0000-0002-5903-6150 (F. Giunchiglia); 0000-0002-1260-4640 (J. Staiano)

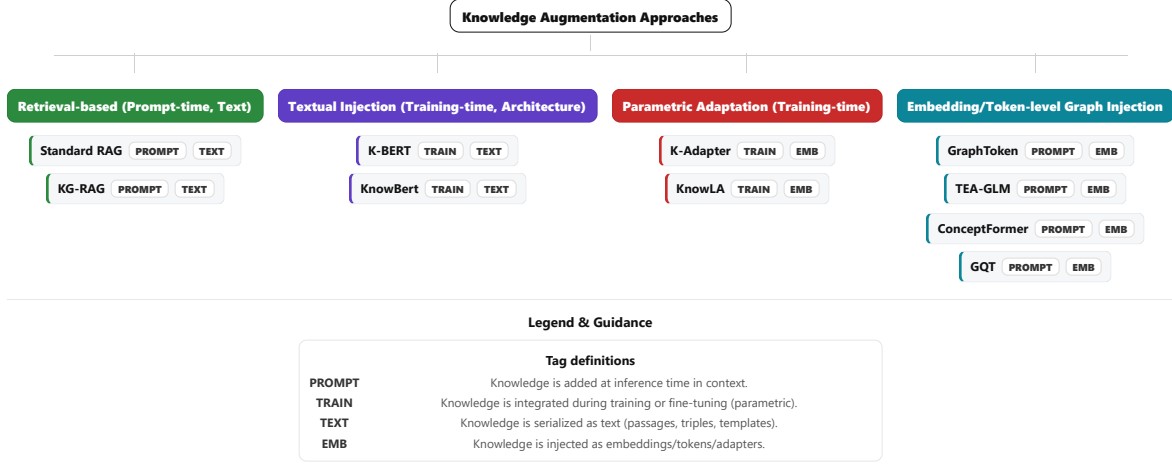

**Figure 1:** Taxonomy of Knowledge Augmentation Approaches for LLMs

research direction for standard graph-related tasks such as node classification and link prediction, while little attention has been given to using this approach to inject **factual** information into LLMs for natural language tasks such as question answering. The few attempts have so far been validated only on small-scale language models [5], leaving open the question of whether embedding-based discrete injection can yield improvements on modern, larger LLM backbones.

In this work, we propose the KoRe architecture, which advances the state-of-the-art by addressing three critical research gaps:

- **Token-Efficient Knowledge Representation:** To overcome the context pressure of text-based injection, KoRe utilizes a Directional Residual Vector Quantization (RVQ) scheme. This compresses graph structures into a minimal sequence of discrete "knowledge tokens," achieving a dramatic reduction in token consumption while preserving the essential factual information required for accurate grounding.
- **Factual Conveyance via GNNs:** Unlike prior GNN-based approaches that focus on standard graph tasks, KoRe investigates whether GNN encodings can effectively transfer the *factual content* of KGs to improve LLM accuracy in question answering.
- **Generalization to unseen entities:** Contrary to previous knowledge integration approaches, our proposed pipeline allows for the encoding of arbitrary knowledge graphs without requiring prior training of KGE models. We use a pretrained sentence-encoder to initialize node and edge embeddings with the intended semantics and let the GNN aggregate this information into a single representation.

## 2. Related Work

To address the knowledge cutoff and factual limitations of LLMs, numerous techniques have been developed to provide models with the information needed to perform open-ended tasks. Some methods focus on the ability to encode knowledge within the model parameters, while others try to embed it into the context window. A taxonomy of the evaluated literature is depicted in Figure 1

Retrieval-Augmented Generation (RAG) [6], along with variants tailored to structured data such as KG-RAG [7], mitigates hallucinations by appending external information to the prompt. While these can be considered elegant and effective solutions, they incur the issue of inflating the context window, ultimately leading to high inference costs. Alternative text-based injection methods, such as K-BERT [8] and KnowBert [9], opt to incorporate knowledge via sentence trees or entity embeddings, but typically require extensive retraining of the LLM backbone or are specialized for a particular set of entities.

Other works have focused on techniques that use Parameter-Efficient Fine-Tuning (PEFT). Methods like K-Adapter [10] and KnowLA [11] use lightweight adapters to incorporate knowledge into the LLM. KnowLA can be considered closest to our proposed approach, as it uses external entity embeddings derived from a Knowledge graph to enhance the LLM. However, it requires embeddings to be pre-computed before training, severely impacting the ability to generalize to unseen graphs.

Another active area of research is the integration of the graph directly as token embeddings, a sort of Graph-Prompting. Examples of this approach are GraphToken [12], TEA-GLM [13], and GQT [14], which use GNNs to generate "soft" token embeddings that represent structural information. While bearing resemblance to our proposed architecture, these methods target graph-specific benchmarks rather than general-purpose factual grounding for LLM reasoning. Further, a more sophisticated integration approach has been proposed in ConceptFormer [5], which injects "concept vectors" to reduce token consumption; while showing promising results for knowledge injection, its evaluation was limited to smaller models like GPT-2 0.1B.

## 3. Methodology

### 3.1. Problem Formulation

Let $G_x = (\mathcal{V}_x, \mathcal{E}_x, \mathcal{R}_x)$ be the 1-hop star subgraph centered on query entity $c$, extracted from Wikidata [15]. Given query $x$ and corresponding $G_x$, the model must generate answer $y$ grounded in the factual triples contained in $G_x$. The task can be seen as a *knowledge-conditioned* language-modelling objective, where the conditioning signal is a compact representation of $G_x$ that fits within the LLM's context.

### 3.2. Model Architecture

KoRe is composed by four trainable modules grafted onto a largely frozen Qwen3-8B backbone with LoRA adaptation (Figure 2):

1. **Graph extraction:** This module uses the relevant entities in the query to extract from WikiData the relevant sub-graph;
2. **Graph Encoding:** First, entities and relations are mapped to dense vectors using a frozen sentence encoder (Qwen-Embeddings); then, a TransformerConv network, with edge-type embeddings and GraphNorm, produces a single $d_{\text{gnn}}$-dimensional graph summary;
3. **Vector Quantization:** The summary is compressed into $Q$ discrete codebook indices $(i_1, \dots, i_Q)$;
4. **Alignment:** The quantised discrete embeddings are projected into the LLM token space, mean-standardisation matching is applied, and a `<KG_EMBEDDING>` placeholder is replaced in the tool-response template.

**Graph Extraction & Encoding**  For each textual instance, we extract relevant knowledge in the form of star graphs centered around key entities mentioned in the text. In our datasets, central entities are pre-annotated. In production scenarios, entity linking tools or LLM-based entity recognition would identify and disambiguate entities before graph extraction. To manage computational costs and focus on the most relevant information, we implement a neighbor selection strategy that ranks entities by their global PageRank scores.

To be fed into a Graph Neural Network, we embed the graph nodes and edges using their labels passed through a frozen sentence encoder $\phi : \text{text} \to \mathbb{R}^{d_\phi}$, where the hidden size $d_\phi$ determines both the node and edge feature dimensions fed to the GNN ($d_\phi = d_{\text{node}} = d_{\text{edge}}$). Using this approach, the initial node and edge representations are already aligned with the text domain, making the mapping to the LLM embedding space easier.

The computed embeddings are fed into a `TransformerConv` GNN layer from [16], followed by `GraphNorm` from [17] and residual connections. This process aggregates the graph information into

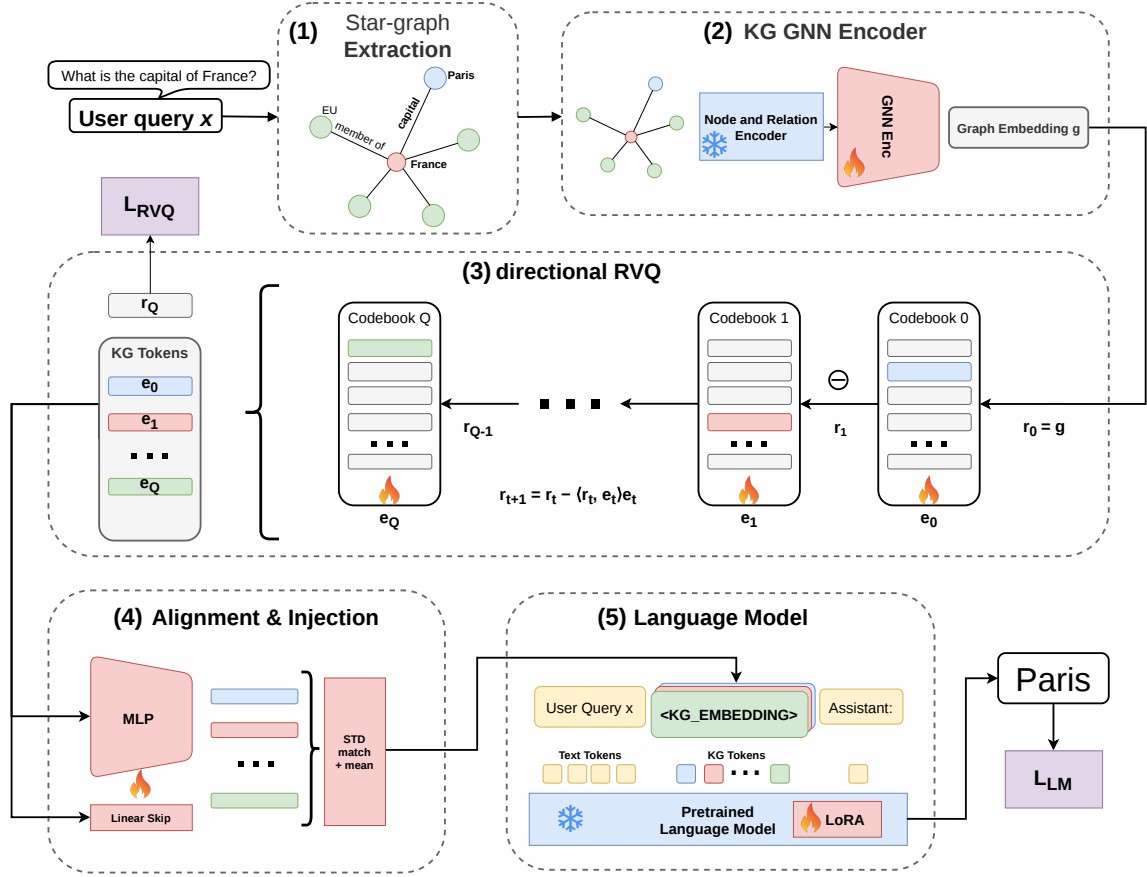

**Figure 2:** KoRe architecture: the query entity is used to extract a star-graph from Wikidata (1), which gets encoded by a TransformerConv GNN with GraphNorm (2), compressed into $Q$ discrete tokens via directional residual vector quantization (3), aligned to the LLM embedding space (4), and injected at the `<KG_EMBEDDING>` placeholder before autoregressive generation (5).

the central node. To obtain the final graph embedding $\mathbf{g} \in \mathbb{R}^d$, we perform central-node pooling by selecting only the representation of the central node. This summary $\mathbf{g}$ is then passed to the residual vector quantization (RVQ) layer to be converted into $Q$ knowledge tokens.

**Directional Residual Vector Quantisation** For compressing the graph representation into multiple discrete vector indices, we follow Meta's GQT [14] and use a residual vector quantization strategy. RVQ iteratively subtracts the chosen code vector from the residual: $r_{t+1} = r_t - e_t$. However, preliminary experiments consistently lead to codebook collapse. To mitigate this issue, we modified the formulation to make use of a $\ell_2$-normalized codebook and a cosine-similarity selection mechanism, as proposed in [18], with a **directional** update:

$$r_{t+1} = r_t - \langle r_t, c_t \rangle \, c_t,$$

which removes only the *component* of the residual along the chosen code direction. This incentivises codes to span orthogonal subspaces, resulting in higher codebook capacity and diversity.

The training loss combines a directional commitment term and a final residual norm:

$$\mathscr{L}_{\text{RVQ}} = \beta \sum_{t=0}^{Q-1} \big( 1 - \cos(r_t, c_t) \big) + \|r_Q\|^2.$$

This, using the Straight-Through Estimator (STE), propagates the gradients to the encoder, ensuring the generated representation uses the codebooks properly. The codes are updated using an exponential

moving average (EMA) of the input residuals with dead-code reset after $N_{dead} * \text{codebook\_size}$ training graphs of unuse (i.e., we require the codes to have a probability of being samples of $1/N_{dead}$).

**Alignment and Injection**  The $Q$ discrete quantized tokens $\mathbf{Z} = [\hat{k}_1; \ldots; \hat{k}_Q]$ are then processed through a residual MLP $f_{\text{out}}$ and a linear skip connection to project these tokens into the LLM's embedding dimension $d_{\text{llm}}$. To ensure the injected tokens are numerically stable, we normalize each graph representation by normalizing its sequence of tokens to a mean of 0 and a variance of 1 across the $Q$ and $d_{\text{llm}}$ dimensions. This normalized tensor is then scaled by the text embedding standard deviation and adjusted by a learned mean shift $\mu_{\text{llm}}$:

$$\tilde{\mathbf{Z}} = \text{StdMatch}(\text{LN}(f_{\text{out}}(\mathbf{Z}) + \text{skip}(\mathbf{Z}))) + \mu_{\text{llm}}$$

The aligned tokens $\tilde{\mathbf{Z}}$ replace a special placeholder token <KG_EMBEDDING> within the prompt, using a prefix mechanism similar to [5].

**Training Objective**  The final composite loss used is

$$\mathscr{L} = \mathscr{L}_{\text{LM}_{(\text{answer tokens only})}} + \mathscr{L}_{\text{RVQ}},$$

where $\mathscr{L}_{\text{LM}}$ is the standard causal language-modelling cross-entropy computed solely on the target answer span. This focus prevents the model from wasting capacity on reconstructing the prompt structure.

## 4. Experimental Design

To train and evaluate our model, we use both synthetic corpora and QA benchmarks.

### 4.1. Datasets

**Tri-REx[19]**  The dedicated dataset Tri-REx Star [20] provides the star graphs extracted from Wikidata for each sentence in Tri-REx. Following the ConceptFormer [5] approach, we use Tri-REx for training our knowledge graph encoder to convey factual information to the LLM backbone. For graph extraction, we reuse the data from Tri-REx star [20] with a maximum of 100 edges. This dataset is well-suited for testing the factual recall ability gained by the model, given the absence of overlap between training, validation, and testing entities, which penalizes models that rely on memorization rather than exploiting the available knowledge.

**SimpleQuestions[21]**  Dataset used to evaluate our model on simple one-hop question answering, and experimenting on continual finetuning. In particular, we use the answerable split provided by [22], which maps the dataset to the Wikidata KG and keeps only the questions for which they were able to find answers in Wikidata by mapping the properties. This leaves the dataset with 14894 train samples, 2210 validation samples, and 4295 test samples. However, due to retrieval limitations from the public SPARQL endpoint[1], we excluded cases where the central or target entity was absent from the retrieved subgraph. This filtering resulted in a final dataset comprising 9294 training samples, 1377 validation samples, and 2677 test samples. For this dataset, in the graph extraction step, we keep up to 10,000 edges, allowing us to evaluate the models under much noisier input conditions.

**WebQSP[23]**  We use this dataset to test the zero-shot capabilities of our model. We never trained on nor used this dataset for validation and held it out only for testing. Similarly to SimpleQuestions, we mapped the dataset to the Wikidata KG using the preprocessed files from [24] and retained up to 10,000 edges during graph extraction.

---

[1]https://query.wikidata.org/bigdata/namespace/wdq/sparql

## 4.2. Baselines

We compare our model against the following baselines:

- **Vanilla Qwen3-8B** (parametric only) to isolate the baseline performance coming from the memorization the LLM underwent during its pretraining.
- **Textualization** (graph triples → natural-language prompt) as the most straightforward injection methodology.
- **LoRA-only** (no KG) to isolate the contribution of our injection mechanism for integrating new knowledge from simply matching the distribution of the training data or memorizing the answers.
- **ConceptFormer** (GPT-2 baseline from literature [5]) to evaluate the scalability of the approach.

## 4.3. Metrics

Our evaluation focuses on the model's ability to correctly predict object entities mentioned in ground truth answers, which are the most critical factual components. For this reason, the primary evaluation metric is **Hit@k**, which measures the proportion of test instances where the correct answer token appears among the top $k$ predictions. For each test instance containing a query $x$ and target answer $y$, we:

1. **Token-level ranking**: For each position $t$ in the target answer sequence, we compute the rank of the true token $y_t$ among all vocabulary tokens based on the model's output logits. The rank is defined as:

$$\text{rank}_t = 1 + \sum_{v \in \mathcal{V}} \mathbf{1}[\text{logit}(v) > \text{logit}(y_t)]$$

   where $\mathcal{V}$ is the vocabulary and $\mathbf{1}[\cdot]$ is the indicator function.

2. **Object boundary identification**: We identify the span of tokens corresponding to the target object entity in the answer sequence, focusing evaluation on factual content rather than auxiliary tokens like articles or prepositions.

3. **Sequence-level rank**: The sequence-level rank is the maximum rank across all object token positions:

$$\text{rank}_{\text{seq}} = \max_{t \in \text{object positions}} \text{rank}_t$$

   This conservative approach ensures that a sequence is considered correct only if *all* object tokens are predicted with high confidence.

We compute Hit@k for $k \in \{1, 3, 5, 10\}$ to capture both strict accuracy (Hit@1) and more lenient retrieval performance (Hit@10). For the WebQSP dataset, as one question can have multiple correct answers, we consider a prediction correct if it matches any of the ground truth answers.

## 4.4. Training Protocol

We evaluate two model configuration checkpoints at different training stages:

1. **KoRe-base** the first model gets trained using only the synthetic sentences from Tri-REx dataset. This step is used as foundation for the GNN and adaptation layers to learn the mapping between knowledge graphs and the LLM token embedding space.

2. **KoRe-QA** then the model is used as base for finetuning, specializing the model on question answering using the SimpleQuestions dataset training split.

**Implementation Details**    We performed multiple ablation studies to determine the optimal configuration for our KG-LM architecture. We analyzed the impact of codebook size, EMA aggressiveness, and quantizer depth (Appendix A). Our results indicate that a moderate codebook size of 128 codes prevents under-utilization while maintaining representational richness. We found that a less aggressive EMA replacement strategy ($N_{dead} = 4$) stabilizes training and improves cross-dataset generalization. Furthermore, increasing the number of quantizers to 20 consistently improved performance on the larger SimpleQuestions graphs compared to Tri-REx, justifying our final choice of Q=20.

Based on these analyses, we fix our final hyperparameters to: codebook size 128, $Q = 20$, EMA dead-code threshold $N_{dead} = 4$, for the RVQ loss $\beta = 0.25$, LoRA is applied to the query, key, value, and output projection matrices with rank $r = 4$ and alpha $\alpha = 8$ with dropout of 0.2. As text encoder for nodes and edge features, we use the Qwen3-Embedding-8B model [25]. While as LLM backbone, we use Qwen3-8B [26]. For training our system, we utilize the AdamW optimizer with a batch size of 8 per GPU (32 total), gradient accumulation steps of 2, gradient clipping with a max norm of 1.0, and validation every 8196 batches for Tri-REx and the entire training set for SimpleQuestions. Given the heterogeneous nature of our model components, we apply distinct learning rates optimized for each parameter group:

- **LoRA parameters**: $1 \times 10^{-5}$ to ensure stable adaptation of the pretrained language model
- **Knowledge Graph Encoder parameters**: $5 \times 10^{-4}$ to enable efficient learning of graph representations

We also employ weight decay at $1 \times 10^{-2}$, the reduce-LR-on-plateau scheduler, using a reduction factor of 0.5 and patience 1, monitoring the Hit@10 on the validation set, and early stopping patience on the same metric of 2.

Each experimental run utilizes:

- **GPU setup**: 4 NVIDIA A100 GPUs with 64GB memory each
- **Training budget**: 8 hours limit per experiment to ensure efficient resource usage.
- **Distributed framework**: We make use of the Accelerate library [27] integrated with DeepSpeed [28] for coordinated multi-GPU training with ZeRO [29] stage 2, which shards the optimizer states across all four GPUs.
- **Memory Optimization**: To reduce the memory footprint and accelerate training, we use the $bf16$ precision option in DeepSpeed.

The distributed training setup enables efficient parallelization of the training for both the language model and knowledge graph encoder components.

## 5. Results and Discussion

This section reports performance across the three knowledge graph question-answering benchmarks we described in Section 4.1: Tri-REx, SimpleQuestions, and WebQSP. WebQSP is a crucial benchmark, as none of our models were exposed to this dataset during training or validation. A summary of the results is shown in Figure 3.

### 5.1. Tri-REx

We first look at the performance on our main training dataset, Tri-REx. Table 1 presents the test set performance, comparing it with our selected baselines. Compared to the base model, our approach hugely improves the performance of the base model. The comparison with **ConceptFormer** reveals competitive performance at Hit@1 (28.1% vs 16.8%) and Hit@10 results (63.5% vs 36.9%); from these initial results, it appears that injecting knowledge vectors scales with model size. In this dataset, the **textualization** baseline outperforms our approach (33.9% vs 28.1% Hit@1). However, our approach's competitive performance demonstrates that quantized representations can effectively capture and utilize

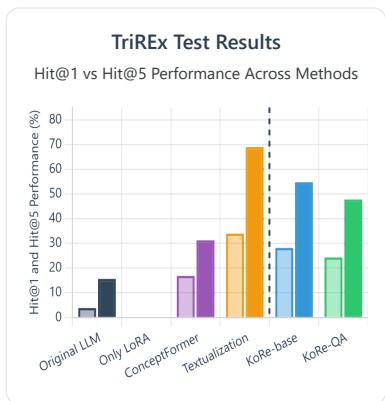 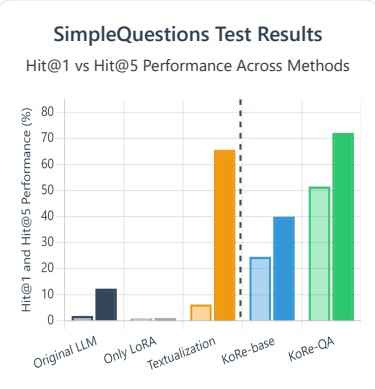 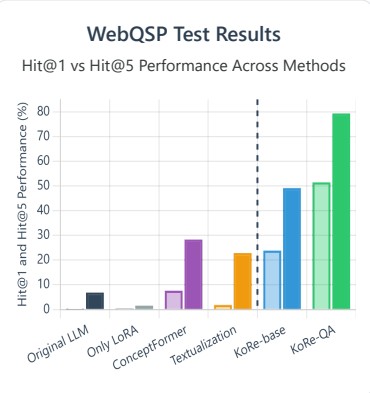

**Figure 3:** Test Results for our baselines and models on the three datasets. Shaded and full bars indicate Hit1 and Hit5, respectively.

factual knowledge. Comparing our model to **LoRA-only** adaptation, we can clearly see how our model successfully generalizes to new knowledge. The tri-REx dataset, as discussed in Section 4.1, is designed to prevent any factual overlap between the training and test sets. Models that rely solely on memorizing the training data facts cannot achieve a good performance on the test set.

**Table 1**

Tri-REx test set results: Performance of best KoRe configurations.

| Configuration | Hit@1 | Hit@3 | Hit@5 | Hit@10 | AvgTokens |
|---|---|---|---|---|---|
| **Baselines** | | | | | |
| Original LLM | 3.8 | 10.4 | 15.6 | 22.3 | 35.9 |
| Textualization | **33.9** | **61.3** | **68.9** | **75.9** | 252.2 |
| Only LoRA (Tri-REx) | 0.0 | 0.0 | 0.0 | 0.2 | 35.9 |
| ConceptFormer (20 CF variant) | 16.8 | – | 31.2 | 36.9 | – |
| **KoRe** | | | | | |
| **KoRe-base** | 28.1 | 46.7 | 54.7 | 63.5 | 70.4 |
| **KoRe-QA** | 24.3 | 40.6 | 47.7 | 56.9 | 70.4 |

## 5.2. SimpleQuestions

The SimpleQuestions test set results demonstrate the power of our discrete knowledge representation approach. Table 2 presents the test set performance, comparing it with our selected baselines. The zero-shot transfer performance (24.5% Hit@1) substantially exceeds the original language model baseline (1.8% Hit@1), confirming that quantized knowledge graphs can effectively convey factual information without task-specific training. The QA fine-tuned model shows even more remarkable performance (51.5% Hit@1), substantially surpassing all the baselines. ConceptFormer is not present in this evaluation as it was not tested on this dataset in the original work.

A notable pattern emerges in the textualization baseline: Hit@1 is only 6.2%, yet Hit@3 recovers sharply to 46.7% and Hit@10 reaches 82.5%. This wide gap is explainable by the exact-match evaluation we use: when presented with serialized triples, the LLM may produce a conversational preamble before stating the answer, penalizing it at rank 1 even though the correct entity is ranked highly. By contrast, KoRe's LoRA fine-tuning on discrete token prefixes may implicitly encourage more direct answer generation, yielding higher Hit@1 scores from the same underlying factual content. We leave a controlled analysis of this generation-style effect to future work.

**Table 2**
SimpleQuestions test set results: Performance of best KoRe configurations.

| Configuration | Hit@1 | Hit@3 | Hit@5 | Hit@10 | AvgTokens |
|---|---|---|---|---|---|
| **Baselines** | | | | | |
| Original LLM | 1.8 | 7.8 | 12.3 | 23.3 | 29.7 |
| Textualization | 6.2 | 46.7 | 65.6 | **82.5** | 248.2 |
| Only LoRA (Tri-REx) | 0.8 | 0.9 | 1.0 | 1.4 | 29.7 |
| **KoRe** | | | | | |
| **KoRe-base** | 24.5 | 36.1 | 39.9 | 46.9 | 64.2 |
| **KoRe-QA** | **51.5** | **67.1** | **72.1** | 77.8 | 64.2 |

## 5.3. WebQSP

The WebQSP evaluation provides valuable insights into the scalability and robustness of our quantized encoding scheme, as our models were never exposed to this dataset during any training stage. Table 3 reports the test set results for the baselines and our model. Even without any exposure to WebQSP during training, the base model achieves a Hit@1 of 23.8%, outperforming all baselines, including a version of ConceptFormer specifically fine-tuned on this dataset. Remarkably, the fine-tuned variant of our model (KoRe-QA) reaches 51.4% Hit@1 and 86.0% Hit@10, more than doubling the accuracy of ConceptFormer and significantly surpassing the textualization baseline. Given that the textualization baseline consumes on average over **10× more tokens** per query, these findings further validate the token-efficient design of our discrete knowledge integration pipeline.

**Table 3**
WebQSP test set results: Performance of best KoRe configurations.

| Configuration | Hit@1 | Hit@3 | Hit@5 | Hit@10 | AvgTokens |
|---|---|---|---|---|---|
| **Baselines** | | | | | |
| Original LLM | 0.1 | 2.7 | 6.8 | 19.2 | 17.2 |
| Textualization | 1.8 | 12.4 | 22.8 | 41.7 | 676.5 |
| Only LoRA (Tri-REx) | 0.4 | 1.2 | 1.5 | 2.0 | 17.2 |
| ConceptFormer (QA fine-tuned on WebQSP) | 7.6 | – | 28.3 | – | – |
| **KoRe** | | | | | |
| **KoRe-base** | 23.8 | 40.7 | 49.1 | 58.5 | 62.9 |
| **KoRe-QA** | **51.4** | **73.2** | **79.3** | **86.0** | 62.9 |

## 6. Conclusion

In this work, we presented KoRe, an architecture for grounding Large Language Models in external Knowledge Graphs without the token overhead of textualization-based approaches. By combining a GNN encoder over 1-hop star subgraphs with a Directional Residual Vector Quantization scheme and a lightweight LoRA adaptation of a frozen Qwen3-8B backbone, KoRe compresses structured factual knowledge into 20 discrete tokens per entity, reducing up to 10× the used tokens compared to serializing the same graph as natural language. We evaluated our models across three benchmarks and demonstrated that compact, discrete knowledge representations can effectively convey factual content to modern LLMs, achieving competitive or superior accuracy to textualization while dramatically reducing context bloating.

## 7. Limitations

Despite the promising results, the proposed approach has several limitations that point to directions for future work. First, the reasoning scope is strictly limited to single-hop: KoRe operates on 1-hop star subgraphs and cannot natively handle multi-hop reasoning, although zero-shot WebQSP results hint at possible extensions towards that. Further, since the datasets used in this work lack reasoning traces, our evaluation is performed with reasoning disabled – the LLM backbone (Qwen3-8B) adopted supports enabling/disabling reasoning ("thinking mode"). We leave the analysis of reasoning impact to future works. In terms of interpretability, tracing which triples influenced a generation is not straightforward: auxiliary decoders could be added to this end in future versions of the architecture. In terms of language coverage, the presented experiments are English-centric; multilingual extensions – which require significant data-collection efforts – are currently ongoing. Finally, our experimental setup currently leverages a single Wikidata KG, we plan to extend the evaluation to unseen knowledge graph schemas.

## Acknowledgments

We acknowledge ISCRA for awarding this project access to the LEONARDO supercomputer, owned by the EuroHPC Joint Undertaking, hosted by CINECA (Italy) .

The work described in this presentation has been conducted within the project TRUMAN. The research leading to these results has received funding from HORIZON-CL4-2024-HUMAN-03, under Grant Agreement no 101214000

Thanks to the developers of ACM consolidated LaTeX styles https://github.com/borisveytsman/acmart and to the developers of Elsevier updated LaTeX templates https://www.ctan.org/tex-archive/macros/latex/contrib/els-cas-templates.

## Declaration on Generative AI

During the preparation of this work, the author(s) used Grammarly to: Grammar and spelling check. Additionally, the OpenWebUI interface with Qwen3.6 and Gemma4 models was used for content enhancement and improving writing style. After using these tool(s)/service(s), the author(s) reviewed and edited the content as needed and take(s) full responsibility for the publication's content.

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

## A. Ablation

To define the best parameters for our architecture, we performed an ablation study on the parameters related to KG compression. These models were trained exclusively on Tri-REx and tested on both Tri-REx and SimpleQuestions validation split.

### A.1. Codebook size

We investigate codebook sizes of {64, 128, 256, 1024} entries while fixing the quantizer depth at $Q$=10 to isolate the impact of vocabulary size. Table 4 and Table 5 report these controlled comparisons. On the original Tri-REx dataset, the model benefited from larger codebooks, but this came at the cost of both codebook utilization and performance on the out-of-distribution SimpleQuestion benchmark. These observations guided our later experiments and led us to choose a codebook size of 128.

**Table 4**
Controlled comparison on Tri-REx (valid): vary Codebook; hold constant Q=10.0, $N_{dead}$=2.

| Codebook Dim | Hit@1 | Hit@3 | Hit@5 | Hit@10 | CodeUtil |
|---|---|---|---|---|---|
| 64 | 30.7 | 48.3 | 55.7 | 63.8 | *99.7* |
| 128 | 31.7 | 48.8 | 56.2 | 64.2 | 78.6 |
| 256 | 31.9 | 49.8 | 57.0 | 65.0 | 49.8 |
| 1024 | *32.5* | *50.1* | *57.5* | *65.4* | 16.6 |

**Table 5**
Controlled comparison on SimpleQuestions (valid): vary Codebook; hold constant Q=10.0, $N_{dead}$=2.

| Codebook Dim | Hit@1 | Hit@3 | Hit@5 | Hit@10 |
|---|---|---|---|---|
| 64 | 10.8 | 26.2 | 32.2 | 40.7 |
| 128 | *28.9* | *43.8* | *49.2* | *56.1* |
| 256 | 18.2 | 35.8 | 41.6 | 47.5 |
| 1024 | 14.3 | 28.1 | 33.0 | 39.7 |

## A.2. EMA Variant

We compare two EMA setups where $N_{dead}$ is set to either 2 or 4. Table 6 and Table 7 report controlled comparisons. The findings indicate that the less aggressive $N_{dead} = 4$ variant achieves better performance and improves code utilization. Relaxing the dead code replacement prevents the quantization process from entering a "thrashing" state, in which codes are repeatedly reassigned without receiving sufficient training signal to stabilize their usage.

**Table 6**
Controlled comparison on Tri-REx (valid): vary $N_{dead}$; hold constant Q=10.0, Codebook=256.0.

| $N_{dead}$ | Hit@1 | Hit@3 | Hit@5 | Hit@10 | Code Util |
|---|---|---|---|---|---|
| 2 | 31.9 | 49.8 | 57.0 | 65.0 | 49.4 |
| 4 | *32.3* | *50.2* | *57.5* | *65.3* | *56.7* |

**Table 7**
Controlled comparison on SimpleQuestions (valid): vary $N_{dead}$; hold constant Q=10.0, Codebook=256.0.

| $N_{dead}$ | Hit@1 | Hit@3 | Hit@5 | Hit@10 |
|---|---|---|---|---|
| $N = 2$ | 18.2 | 35.8 | *41.6* | *54.29* |
| $N = 4$ | *24.0* | *36.1* | 41.0 | 48.7 |

## A.3. Number of Quantizers

The quantization depth directly affects both how expressive our knowledge graph encodings are and how many tokens are needed for knowledge injection. We experiment with $Q \in \{5, 10, 20\}$ quantizers. Table 8 and Table 9 report the controlled comparisons. We observe that increasing quantization depth enhances model performance. However, it also exhibits diminishing returns, suggesting that, under our current architecture, additional quantizers add little useful signal. This is probably a consequence of our simple strategy for selecting the final graph representation; more sophisticated approaches might better leverage the finer-grained reconstructions provided by multiple quantizers.

**Table 8**
Controlled comparison on Tri-REx (valid): vary Q; hold constant Codebook=256.0, $N_{dead}$=2.

| Q | Hit@1 | Hit@3 | Hit@5 | Hit@10 | AvgTokens |
|---|---|---|---|---|---|
| 5.0 | 30.9 | 48.4 | 55.7 | 63.7 | 56.30 |
| 10.0 | *31.9* | *49.8* | *57.0* | *65.0* | 61.30 |
| 20.0 | 31.2 | 48.7 | 55.8 | 63.9 | 71.30 |

**Table 9**
Controlled comparison on SimpleQuestions (valid): vary Q; hold constant Codebook=256.0, $N_{dead}$=2.

| Q | Hit@1 | Hit@3 | Hit@5 | Hit@10 | AvgTokens |
|---|---|---|---|---|---|
| 5.0 | 23.0 | 32.7 | 38.4 | 46.1 | 49.27 |
| 10.0 | 18.2 | 35.8 | 41.6 | 47.5 | 54.29 |
| 20.0 | *28.2* | *42.0* | *47.4* | *54.9* | 64.32 |