# OpenReview forum: "KoRe: Compact Knowledge Representations for Large Language Models"
_ijcai.org/IJCAI-ECAI/2026/Workshop/GENAIK-NORA — IJCAI-ECAI 2026 Joint Workshop on GENAIK and NORA_

### Official Review · Reviewer_9TmL · 2026-05-27
**This paper presents a system that injects knowledge graph into LLM’s generation, which demonstrates a certain degree of innovation, generalization and reproducibility. However, it may perform averagely in some domains and the paper itself could benefit from improved readability.**

**Rating:** 7
**Confidence:** 4

**Review:**

Quality
The paper provides a very detailed explanation of the solutions to the hallucination problems of current LLMs and also discusses their issues. Based on this, the paper proposes a system that injects knowledge graphs into large models in the form of embedding representation. The Directional Residual Vector Quantisation mentioned in the paper is used to solve the problem of codebook collapse, which is quite impressive. By combining with other effective methods, this system ultimately demonstrated its effectiveness on some reliable datasets. And there are also reliable analytical results as well as relevant limitations. In addition, the paper also provides the GitHub link, which enhances the reproducibility of the system.
Clarity
The overall logic of the paper is clear, but in the Model Architecture section, more detailed content needs to be added to enhance readability.
Originality
This paper has referred to many other related papers and has built its own system. In some details, it also demonstrates a certain degree of originality.
Significance
The paper aims to address the hallucination problem that is of great significance for LLMs. From this perspective, this paper is meaningful. However, although this system has outstanding generalization ability, its performance within specific domains still needs to be further improved.
Pros and Cons
The paper has good quality, clarity, originality and significance mentioned above. However, in some details, the paper needs to improve its readability:
1. In the Introduction, it states "by addressing four critical research gaps" but only mentions three of them.
2. In the Graph Extraction & Encoding section of 3.2, it directly presents the edges of Tri-REx and QA datasets, which may cause confusion.
3. Figure 2 is too simple and some names are not labeled, making it difficult to correspond with the text.

---

### Official Review · Reviewer_fgrB · 2026-06-02
**This paper introduces a method to encode 1-hop sub-graphs into compact discrete knowledge tokens and inject them into an LLM backbone.**

**Rating:** 7
**Confidence:** 4

**Review:**

The paper presents a well-structured and comprehensive study addressing a critical limitation in modern LLMs: their implicit, opaque knowledge encoding which hinders transparency, updatability, and factual accuracy.

The proposed method is grounded on solid principles combining Graph Neural Networks (GNNs) for knowledge graph encoding with Directional Residual Vector Quantization (RVQ) to achieve compact discrete token representations.

The writing is clear and concise, with a logical flow from motivation through related work to methodology and experiments.

Pros
- It uses Directional RVQ to compress one-hop subgraphs into ~20 discrete tokens, significantly cutting down token usage compared to text-based knowledge injection (e.g., 252 tokens vs. 20 tokens).
- It achieves superior or competitive accuracy on three QA benchmarks, including zero-shot and fine-tuned settings.
- It encodes arbitrary knowledge graphs using pretrained sentence embeddings and GNN aggregation, avoiding costly KGE precomputation.

Cons
- Focus on 1-hop star subgraphs inherently limits reasoning to immediate relations, excluding multi-hop or complex inference chains.
- The adopted LLM backbone's "thinking mode" (reasoning) was disabled during experiments, leaving open questions about synergy with KoRe and reasoning.
- Although knowledge is represented discretely, tracing which specific triples influenced model outputs is not straightforward.
- Experiments are in English using Wikidata subsets; performance on multilingual or other domain-specific knowledge graphs remains untested.

---

### Official Review · Reviewer_Cnvq · 2026-06-05
**Nice work using VQ for small 1-hop knowledge graphs**

**Rating:** 8
**Confidence:** 5

**Review:**

This paper proposes encoding 1-hop Wikidata subgraphs into compact discrete tokens via a GNN encoder followed by Residual Vector Quantization, injecting the result into a frozen Qwen3-8B backbone with LoRA adaptation. The stated goals are token efficiency (up to 10x reduction vs. textualization) and generalization to unseen graphs without KGE pretraining.

The generalization to unseen graph is a good result obtained by this work, with a practical advantage over similar recent works for KG completion like ReaLM (Guo et al., 2026) and KoPA (Zhang et al., 2024), which require pretraining KGE models on the target graph.

The directional RVQ formulation (projecting out only the residual component along the chosen code direction) is a nice small improvement over the original VQ proposal.

Nevertheless I have some remarks:

Since the textualization baseline suffers from the presence of the conversational preamble (as acknowledged by the authors), and the evaluation follows similar retrieval-oriented evaluation measures, I think that measures such as MRR (Mean Reciprocal Rank) or nDCG would be fairer for the comparison to the baselines.

Second, the identification of the "central entity" in the query and its span is a bit vague. It may be unambiguously identified in the datasets used for evaluation, but what would happen in a realistic scenario? How the entity would be identified and disambiguated?

The authors mention filtering cases where the central entity was absent from the retrieved subgraph, which reduced SimpleQuestions from 14k to 9k samples. It's a 37% data loss that may hamper reproducibility efforts: the authors should publish their dataset if someone had to work on the same set of questions and compare with them.

---

### Official Review · Reviewer_Vucm · 2026-06-06

**Rating:** 6
**Confidence:** 3

**Review:**

KoRe encodes Wikidata subgraphs via a GNN and compresses the embeddings into discrete “knowledge tokens” using Directional RVQ, injecting them into a frozen Qwen3-8B with LoRA adapters. Evaluated on Tri-REx, SimpleQuestions, and WebQSP, it matches or outperforms textualization baselines while cutting context length by up to 10 times.

Strengths

- The technical pipeline is well-conceived and addresses a real bottleneck of token inefficiency in LLM–KG integration.
- The practical results show a 10 times improvement in token efficiency.
-  The zero-shot WebQSP results suggest effective knowledge representation transfer across QA domains.

Weaknesses
- The paper does not provide an ablation study.
- The mean-standardisation matching step in the alignment module could benefit from a more intuitive explanation or a small running example.
- The ConceptFormer baseline uses a GPT-2 0.1B backbone, making direct performance comparisons with KoRe’s Qwen3-8B misleading. A same-backbone ablation would be more informative.
- Reported metrics lack confidence intervals, p-values, or variance across random seeds, making it hard to judge whether differences are statistically robust.
- Restricted to 1-hop star subgraphs; multi-hop reasoning, which is essential for complex QA, is not addressed natively.

---

### Decision · Program_Chairs · 2026-06-10

Accept